# Oxygen Concentration Plays a Critical Role in Fibrinogen-Mediated Platelet Activation via Inactivation of α_IIb_β_3_ and Modulation of Fibrinogen

**DOI:** 10.3390/biom15040501

**Published:** 2025-03-29

**Authors:** Sophie V. L. Leonard, Zoe Booth, Leigh Naylor-Adamson, Lewis Bibby, Katie S. Wraith, Giordano Pula, Monica Arman, Simon D. J. Calaminus

**Affiliations:** Biomedical Institute for Multimorbidity, Hull York Medical School, University of Hull, Hull HU6 7RX, UK

**Keywords:** platelet, actin cytoskeleton, fibrinogen, oxygen, hypoxia, integrin function

## Abstract

In the vascular system, pathological conditions that cause hypoxia are associated with increased platelet activity and thrombosis. Using a platelet spreading assay, we show that severe hypoxia (i.e., 1%), venous (i.e., 5%), and, surprisingly, arterial (i.e., 12%) oxygen concentrations cause a significant reduction in platelet surface area coverage on fibrinogen in comparison to atmospheric oxygen condition (i.e., 21% oxygen), whilst adhesion and spreading on collagen and CRP were not affected. Importantly, the addition of thrombin or zinc restored full platelet spreading on fibrinogen, indicating that the inhibition of platelet spreading on fibrinogen was due to defective integrin activation. Analysis of integrin activation with FACs via PAC-1 staining supported a significant reduction in integrin activation in hypoxic conditions. Interestingly, a fibrinogen matrix prepared at 1%, 5%, or 12% oxygen failed to induce full platelet spreading, even when the experiments were performed at atmospheric oxygen concentration, indicating that the structure and activity of the fibrinogen coating is affected by oxygen. The effect of oxygen on different matrix proteins is critical to understand, as these data clearly demonstrate that collagen and CRP can support platelet activation at all O_2_ concentrations, whilst fibrinogen mediated platelet activation and spreading is lost at physiological and pathological O_2_ concentrations. These data have clear implications for thrombus formation data and highlight the role of oxygen in regulating platelet function.

## 1. Introduction

Platelets are small (2–3 μM) anucleate blood cells derived from megakaryocytes that have an average lifespan of 8 to 10 days [1]. Platelets play an essential role in haemostasis, binding to a site of injury and forming a platelet plug to prevent excessive bleeding [2]. Upon vascular injury, the endothelial layer is disrupted and extracellular matrix (ECM) proteins, such as collagen, von Willebrand factor (vWF), and fibrinogen, are exposed to platelets [3]. Platelets bind to the ECM proteins via specific receptors, such as GPIb (vWF), GPVI, and α_2_β_1_ (collagen), and integrin α_IIb_β_3_ (fibrinogen) [4,5,6,7]. The binding of agonists to these receptors activates multiple intracellular signalling cascades to initiate the secretion of dense and alpha granules. These granules store an abundance of platelet agonists, such as zinc, adenosine diphosphate (ADP), and serotonin, which are secreted into the bloodstream causing the activation of surrounding platelets, ultimately aiding thrombus formation [8,9,10].

As platelets move around the body via the bloodstream, they are exposed to different levels of oxygen (O_2_) [11]. The arteries transport oxygenated blood (approximately 12% O_2_) at high pressure and volume from the lungs to the heart, which then pumps it into the systemic circulation. The veins, on the other hand, carry slow-moving deoxygenated blood (approximately 5% O_2_) at low pressure back to the lungs [12,13]. However, in certain diseases, such as atherothrombosis, stroke, chronic obstructive pulmonary disease (COPD), and sleep apnoea, tissue hypoxia (approximately 1% O_2_) can occur [14,15,16,17]. The presence of hypoxia has been linked to hyperactive platelets and an increased thrombotic risk in both human and in vivo models [18,19,20,21]. However, the underlying mechanism of this increase in hyperactivity is unclear, with some studies suggesting that healthy platelets exposed to low O_2_ have upregulated expression of membrane P-selectin and activated integrin α_IIb_β_3_ expression in response to agonist stimulation [22], whilst others suggest that α_IIb_β_3_ activity is markedly reduced following platelet tissue exposure to hypoxic conditions [23]. Part of the issue is the variation in O_2_ concentration used and the lack of consistency within the experimental approach taken. We have previously shown how quickly platelets experience hypoxia is dependent on the volume and concentration of platelets present [24]. Control of these factors is critical to ensuring the correct O_2_ level and consistency of platelet response at different O_2_ levels [24].

Therefore, to fully investigate the effect of variable O_2_ concentrations on platelet spreading, we spread washed platelets on a range of different ECM proteins at 1% (hypoxic), 5% (venous), 12% (arterial), and 21% (normoxic) conditions and monitored how the platelets responded. This allowed us to establish how platelets behaved in a range of relevant O_2_ conditions and therefore to identify if any changes to the actin cytoskeleton induced in different O_2_ conditions could underpin why low O_2_ conditions encourage a prothrombotic phenotype.

## 2. Materials and Methods

### 2.1. Reagents

Chrono-Par collagen suspension (Chrono-Log Corporation, Havertown, PA, USA), fibrinogen (Enzyme Research, Swansea, UK), fibronectin (ThermoFisher Scientific, Loughborough, Leicestershire, UK), collagen-related peptide (CRP-A) (PPlus Medical, Letterkenny, Donegal, Ireland), alexa-fluor 647 labelled fibrinogen (ThermoFisher Scientific, Loughborough, Leicestershire, UK), and Eptifibatide (Tocris Bioscience, Abingdon, UK). ProLong diamond antifade mountant and paraformaldehyde (ThermoFisher Scientific, Loughborough, Leicestershire, UK). PPACK (D-Pehnylalanyl-L-prolyl-L-arginine chloromethyl ketone) and Fluoroescein Isothiocyanate (FITC) phalloidin (Enzo Life Sciences, Farmingdale, NY, USA). Phosphate-buffered solution (PBS) tablets (Gibco, Paisley, Scotland). CD42b Brilliant Violet and PAC1 FITC antibodies (BioLegend, San Diego, CA, USA). All other chemicals were from Sigma Ltd. (Poole, UK) unless otherwise stated.

### 2.2. Ethics and Donor Recruitment

Work was undertaken in accordance with NHS REC study 21/SC/0215 ‘Investigation of Blood cells for research into Cardiovascular disease’. Blood was obtained from drug-free, healthy volunteers who provided written informed consent.

### 2.3. Preparation of Blood

All experiments were conducted within 4 h of blood collection. For washed platelets (WP), whole blood was collected into acid citrate dextrose (ACD-A) and then centrifuged at 60× *g* for 10 min. Plasma was removed, and the blood was respun at 100× *g* for a further 10 min. The plasma was removed and added to the previously collected platelet-rich plasma (PRP). The PRP was treated with citric acid (0.3 mM) and centrifuged at 1000× *g* for 10 min. The platelet pellet was then resuspended in wash buffer (36 mM citric acid, 10 mM EDTA, 5 mM glucose, 5 mM KCl, and 9 mM NaCl) and centrifuged for a further 10 min at 1000× *g*. Platelets were counted using a Beckman Z1 Coulter Particle Counter (Beckman Coulter, High Wycombe, UK) and resuspended to 2.5 × 10^8^ platelets/mL using modified Tyrode’s buffer (20 mM HEPES, 134 mM NaCl, 2 mM KCl, 0.34 mM Na_2_HPO_4_, 5.6 mM d-glucose anhydrase, 12 mM Na_2_CO_3_, and 1 mM MgCl_2_).

### 2.4. Deoxygenation of Platelets

The rate of platelet deoxygenation was established across a range of platelet concentrations using an OXY-4 ST oxygen meter attached to a Needle-Type Oxygen Microsensor (NTH-PSt7) (PreSens, Regensburg, Germany). WP at 2.5 × 10^8^ platelets/mL were moved into a Whitley H35 Hypoxystation (Don Whitley Scientific, Bingley, UK) (set at either 1%, 5%, or 12% O_2_) and left for 2 h at 37 °C to reach equilibrium with the chamber. The corresponding normoxic control for each donor was also left in an incubator for 2 h at 37 °C under standard laboratory O_2_ levels/conditions. Following the 2-h incubations, platelets were diluted to the desired concentration using Tyrode’s buffer at the equivalent O_2_ concentration.

### 2.5. Platelet Spreading

Coverslips were coated with either collagen (100 μg/mL), CRP-A (100 μg/mL), fibrinogen (3 μg/mL or 100 μg/mL), or fibronectin (100 μg/mL). To allow enough time for the matrix to fully adhere/develop, the 3 ug/mL fibrinogen coverslips were coated overnight at 37 °C, while the 100 μg/mL fibrinogen, collagen, CRP-A, and fibronectin coverslips were coated for 1 h at 37 °C [25,26,27,28,29]. Fibrin coverslips were coated with fibrinogen (100 μg/mL) for 1 h before treatment with thrombin (1 U/mL) for 15 min and blocked with PPACK (40 μΜ) for 30 min [30]. The coverslips were washed twice with PBS between each step. Coverslips were then coated with denatured bovine serum albumin (BSA) (5 mg/mL) for 1 h at 37 °C. Coverslips were washed twice with PBS before the addition of WP (2 × 10^7^ platelets/mL) to the coverslips for 25 min at 37 °C in the presence or absence of zinc sulfate (100 μM, zinc), thrombin (0.1 U/mL), or eptifibatide (9 mM). Coverslips were washed with PBS and fixed with 4% paraformaldehyde (PFA) for 10 min. Coverslips were then removed from the hypoxia chamber and permeabilized with 0.1% Triton X-100 for 5 min. Coverslips were stained with FITC-phalloidin for 1 h at room temperature before being washed with PBS and mounted with ProLong Diamond Antifade Mountant. Coverslips were imaged on a Zeiss Axio Imager fluorescence microscope with an ×63 oil immersion objective (1.4 NA) using Zen Pro software (blue edition) v2.0.0.0 (Zeiss, Cambridge, UK). Five images were captured per condition. Platelet adhesion and the platelet surface area were analyzed using ImageJ v1.54d (National Institutes of Health, Bethesda, MD, USA). Blinded imaging and analysis was carried out for Figures 3, S2 and S5.

### 2.6. Reoxygenation of Platelets

Following the 2-h deoxygenation in the hypoxia chamber, WPs (2.5 × 10^8^ platelets/mL) were removed from the hypoxia chamber and left to reoxygenate under stirring conditions for 5 min before dilution to the desired concentration using normoxic Tyrode’s buffer.

### 2.7. Characterization of Fibrinogen Surface

Coverslips were coated with fluorescently labelled fibrinogen (100 μg/mL) for 1 h before being washed with PBS, blocked with denatured BSA (5 mg/mL) for a subsequent hour, and then fixed with 4% PFA for 15 min. Coverslips were then removed from the hypoxia chamber. Coverslips were mounted with ProLong Diamond Antifade Mountant (ThermoFisher Scientific, Loughborough, Leicestershire, UK). Coverslips were imaged on a Zeiss Axio Imager fluorescence microscope with an ×63 oil immersion objective (1.4 NA) using Zen Pro software (Zeiss, Cambridge, UK). Five images were captured per condition. All coverslips were kept in the dark throughout the experiment. 

### 2.8. Flow Cytometry

WPs (1 × 10^7^ platelets/mL) were stained with platelet-specific CD42b Brilliant Violet and PAC1 FITC antibody and treated with 0.1 U/mL thrombin for 20 min. Following stimulation, platelets were fixed with 0.5% PFA and analyzed using a BD LSR Fortessa cell analyzer (BD Bioscience, Franklin Lakes, NJ, USA), with a valid CS&T procedure completed each experimental day. For each sample, 10,000 CD42b specific events were read by the FACSDiva Software v9.7 (BD Biosciences, Franklin Lakes, NJ, USA) and then were analyzed using Floreada.io (https://floreada.io (accessed on 23 January 2025)).

### 2.9. Statistical Analysis

All statistical tests were performed using GraphPad Prism 8.0.1 software (GraphPad, San Diego, CA, USA). Data are presented as mean ± standard error of the mean (SEM), unless otherwise stated. Data were analyzed by paired *t*-tests and a one-way or two-way ANOVA, followed by a post-hoc test of Tukey’s multiple comparison. *p* < 0.05 was considered significant.

## 3. Results

### 3.1. Identification of the Correct O_2_ Concentration

The vast majority of in vitro experiments are performed at 21% O_2_ [24]. However, this O_2_ concentration is not found anywhere in the human body, so it is not physiologically relevant, and therefore it is crucial to identify how physiological O_2_ levels are relevant to platelet function [11]. To complete this, we first identified the correct procedure to incubate the platelets at the relevant O_2_ concentrations.

Initially, we confirmed the length of time taken for platelets to experience 1% O_2_ over a 3 h period. Concurrently, we monitored that platelets stored at 21% did not experience reduced O_2_ while incubated at 37 °C. When incubated at 1% O_2_, within 30 min, the O_2_ concentration in a solution of 2.5 × 10^8^/mL platelets was at equilibrium with the chamber (Appendix A). Therefore, platelets were incubated for 2 h prior to use to ensure that they experienced a sustained period of the relevant O_2_ concentration before use within the spreading assay.

### 3.2. Platelet Spreading Is Impacted by O_2_ Concentration

After identifying the relevant incubation time for O_2_ equilibration, we next investigated if the various O_2_ concentrations induced changes in platelet spreading. Therefore, platelets were incubated at different O_2_ concentrations before being spread on either collagen, CRP-A, fibrin, fibronectin, low-density fibrinogen, or high-density fibrinogen for 25 min before fixation, staining, and imaging.

Analysis of our data showed that both platelet adhesion and the average surface area of platelets spread on collagen were unaffected at all O_2_ concentrations tested compared to the corresponding 21% O_2_ control (Figure 1). In addition, platelets spread on CRP-A at 1% O_2_ were also unchanged to those in a 21% O_2_ control (Appendix A).

In contrast, although platelet adhesion was unaffected, platelets spread on 100 μg/mL fibrinogen at 1%, 5%, and 12% O_2_ showed a significant reduction of 16.34 µm^2^, 13.43 µm^2^, and 18.91 µm^2^, respectively, in the average surface area of platelets spread on fibrinogen in comparison to their normoxic control (Figure 2). Similar results were observed when platelets were spread on 3 μg/mL fibrinogen, whilst platelets did not spread well on fibronectin at any of the O_2_ concentrations tested (Appendix A). Moreover, we also showed no difference in the level of platelet spreading on fibrinogen (100 μg/mL) in the presence of the α_IIb_β_3_ inhibitor, eptifibatide, at 1% O_2_ (Appendix A). Our findings suggest that while integrin α_IIb_β_3_ can still interact with and mediate platelet adhesion to the fibrinogen matrix, it fails to promote platelet spreading and activation, while GPVI-mediated platelet spreading is unaffected by changes in O_2_ concentration.

After identifying the influence of O_2_ concentration on a fibrinogen matrix compared to collagen, we sought to spread platelets on fibrin (100 μg/mL) (Figure 3). While adhesion to the matrix was unaffected by O_2_ concentration, platelets kept at 1% O_2_ showed a clear reduction (13.37 μm^2^) in surface area when compared to their normoxic counterpart. Moreover, in the presence of eptifibatide, the platelet spreading was further reduced. These data suggest that fibrin is at least partially able to induce spreading, although the impact of reduced O_2_ on integrin α_IIb_β_3_ function still impedes the level of spreading.

### 3.3. Platelet Spreading Is Restored by Addition of Thrombin or Zinc at All O_2_ Concentrations

We then sought to establish whether the lack of spreading on a fibrinogen matrix at lower O_2_ concentrations was due to a lack of overall integrin α_IIb_β_3_ activity or a lack of response to the fibrinogen. To assess this, we treated platelets with either zinc or thrombin before spreading on fibrinogen. Zinc and thrombin are both prime examples of direct platelet agonists and have previously been shown to induce full platelet spreading on a fibrinogen matrix [25,31].

Treating platelets with either zinc or thrombin before spreading re-established platelet spreading across all of the O_2_ concentrations tested (including normoxia). We observed a significant increase of 20.15 µm^2^ and 17.37 µm^2^ in the platelet surface area following zinc and thrombin treatment, respectively, when compared to the control in hypoxia (Figure 4). Platelets responded to zinc and thrombin in a similar manner when spread on 3 μg/mL fibrinogen and 100 μg/mL fibronectin (Appendix A) in a hypoxic environment, as well as at arterial and venous O_2_ levels (Appendix A). Our data suggest that platelet function per se is not altered by low O_2_ concentrations, but platelet spreading on fibrinogen and fibrin is specifically reduced.

To further examine the level of integrin α_IIb_β_3_ activity, we incubated washed platelets at 1% and 21% O_2_ before stimulation with thrombin. We then analyzed the level of integrin activation using a PAC-1 antibody via flow cytometry. This data identified that whilst thrombin induced a strong activation of integrin α_IIb_β_3_ in normoxic conditions, at lower O_2_ concentrations, the integrin was poorly activated (Appendix A).

### 3.4. Preparation of Fibrinogen in Different O_2_ Concentrations Alters Platelet Spreading

Although the lack of spreading on fibrinogen is potentially due to a lack of integrin activation, we also sought to investigate if the reduction in fibrinogen-mediated platelet spreading observed at physiologically relevant O_2_ levels could be attributed to the fibrinogen matrix itself. Therefore, fibrinogen was coated on coverslips in 1%, 5%, 12%, and 21% O_2_ conditions before the addition of platelets either at 1%, 5%, 12%, or 21% O_2_.

Analysis of the data showed a clear difference in the level of platelet spreading between the four conditions investigated (Figure 5). Platelets spread solely in the normoxic environment had the largest average surface area (30.76 µm^2^). As expected, fibrinogen prepared at 21% O_2_ did not support full spreading if the platelets were incubated at 1% O_2_. Interestingly, normoxic platelets spread on fibrinogen prepared in 1% O_2_ conditions were also significantly inhibited, suggesting that the fibrinogen matrix itself was no longer supporting effective platelet spreading. These results were replicated at 5% O_2_ (Appendix A). At 12% O_2_, although the trend was still present, the results were no longer significantly different (Appendix A).

To understand if platelet spreading on fibrinogen could be recovered, we reoxygenated washed platelets stored at 1% O_2_ by stirring them for 5 min using a light transmission aggregometer in an open cuvette at 21% O_2_ before spreading them on fibrinogen prepared both at 21% and 1% O_2_. Analysis of this data identified that the reoxygenated platelets responded as expected and completed a similar level of spreading to that seen with platelets at 21% O_2_. However, they were still unable to spread on fibrinogen prepared at 1% O_2_, indicating that the fibrinogen matrix was still unable to induce platelet activation and spreading (Appendix A). To ensure it was not due to an alteration in fibrinogen coverage in hypoxic conditions, alexa-fluor 647-fibrinogen was coated onto coverslips and then imaged. This identified a similar level of fibrinogen coverage at both 21% and 1% O_2_ (Appendix A).

Our data indicate that both platelets themselves and the fibrinogen matrix are impacted by a reduction in O_2_ level to a hypoxic level.

## 4. Discussion

Globally, cardiovascular disease (CVD) is the leading cause of mortality, with an estimated 17.9 million deaths in 2019 (Cardiovascular Diseases (CVD), available online https://www.who.int/news-room/fact-sheets/detail/cardiovascular-diseases-(cvds) (accessed 25 September 2024)) [32]. With an aging population and increasing numbers of obese and diabetic patients, this figure is predicted to grow further [33]. Key to this disease is the role of platelet hyperreactivity, but also periods of reduced O_2_ levels within the vascular system, both of which have been noted to provide a prothrombotic environment [14,34,35,36]. Thus, it is important to understand the influence O_2_ concentration might have on platelet activation.

Here, we used the platelet spreading assay to present evidence that: (1) Only washed platelets kept in a normoxic environment have sustained integrin α_IIb_β_3_-mediated platelet spreading. (2) Collagen and CRP spreading is unaffected by O_2_ concentration. (3) The inhibition of platelet spreading at 1%, 5%, and 12% O_2_ is removed through the addition of thrombin and zinc. (4) Fibrinogen itself is less reactive if prepared in low O_2_ conditions, indicating that O_2_ can also influence the matrix protein fibrinogen’s function. (5) Reoxygenation of platelets restores platelet spreading, but these platelets still cannot spread on fibrinogen prepared at 1% O_2_. Our data therefore indicates that O_2_ can affect platelet activation by modulating integrin activation in washed platelet conditions, but also critically by reducing the ability of fibrinogen itself to activate platelets directly. These data underscore the importance of using physiologically relevant O_2_ conditions within experimental conditions.

The platelet spreading assay has been used to show how platelets respond to different environmental conditions, including O_2_ concentrations [23,24,37,38]. Here, we showed that platelets can adhere successfully to all matrices tested at each O_2_ concentration but spread significantly less at 1%, 5%, and 12% O_2_ in comparison to the normoxic control on fibrin, fibronectin, and fibrinogen. This work agrees with previous publications but importantly deepens our understanding of O_2_’s role in platelet spreading by utilizing a wider range of O_2_ concentrations and matrix proteins than other publications. Importantly, we demonstrated that the reduction in platelet surface area and spreading on fibrinogen observed by Kiouptsi et al. [23] is not a phenomenon observed only in hypoxia but is also present at physiologically relevant 5% and 12% O_2_. This indicates that the normoxic condition is the outlier, as it allows full spreading in comparison to the more physiologically relevant 1%, 5%, and 12% O_2_ concentrations. However, the absence of a response to the fibrinogen matrix at 1%, 5%, and 12% O_2_ is not due to a lack of the ability of the platelets to activate, as the addition of thrombin or zinc induced full platelet spreading. Thrombin and zinc are both known platelet agonists, with zinc stored within alpha granules and secreted by platelets potentiating platelet activation [39]. Platelet incubation with zinc and thrombin can induce platelet activation through secondary pathways to integrin α_IIb_β_3_ [6,39,40]_._ Whilst our data show that platelets are able to spread on the fibrinogen matrix at low O_2_ concentrations in the presence of an agonist, this is likely due to inside-out signalling induced by zinc or thrombin, rather than outside-in signalling by fibrinogen. Therefore, the activation of the platelet by the fibrinogen itself is circumvented, allowing the platelet to fully spread.

Interestingly, this reduction in spreading seems to be a combination of two effects. Firstly, the integrin is activated to a lesser extent in lower O_2_ concentrations compared to 21%. Although this is as previously published for 1%, the 12% O_2_ data is surprising [23]. The integrin activation seems to be much more sensitive to O_2_ concentration than expected. In addition to the reduction in integrin activation, we also established that the ability of fibrinogen to activate platelets was affected by the level of O_2_, as platelets at 21% O_2_ can fully activate α_IIb_β_3_ and yet were unable to spread on fibrinogen prepared at 1%, 5%, or 12% O_2_. Furthermore, reoxygenated platelets recovered their integrin function, in agreement with Kusanto et al. [24], and Kiouptsi et al. [23], and showed full spreading on fibrinogen at 21% O_2_, but were still unable to spread on the fibrinogen matrix prepared at 1% O_2_. Although it is unclear why there is a reduction due to the preparation of fibrinogen at 1%, we were able to show that fluorescently labelled fibrinogen covered the coverslip in similar manners at both 21 and 1%, indicating this aspect is not affected.

This reduction in the reactivity of the fibrinogen matrix is therefore critical to understand, as it asks key questions about the role of fibrinogen within thrombi, as in vivo thrombi will form at between 5–12% O_2_ depending on the location of the thrombus. Firstly, is fibrinogen in the blood similarly affected? Secondly, how relevant is platelet spreading on fibrinogen, or is fibrinogen only required for platelet adhesion? These two questions are key, as the answers would alter our understanding of the role of fibrinogen within a thrombus. There are hints at the answers, as Schurr et al. identified, for example, that lamellipodia are not required for thrombus formation in vivo [41], indicating that the adhesion we see here on fibrinogen at 1% O_2_ could still support thrombus formation. It is also important to note that the platelet response to collagen is unaffected by O_2_ conditions and therefore can still act as a matrix protein capable of both adhering and activating platelets, demonstrating its critical prothrombotic nature. Overall, the effect of O_2_ on fibrinogen (and other matrix proteins) is certainly an area for further research.

It is important to acknowledge, however, that although the hypoxia chamber is an effective mechanism to induce a constant O_2_ environment, it still has some limitations when replicating O_2_ conditions experienced in vivo. Most importantly, O_2_ gradients that are found in vivo are not possible to replicate within the chamber [42]. Secondly, the time frame for blood cells to experience different O_2_ concentrations may not accurately replicate that which happens in the body.

## 5. Conclusions

Overall, our data identify that the ability of platelets to spread on fibrinogen in washed platelet conditions is defined by two key factors: the activity of the integrin α_IIb_β_3_ and the preparation of the fibrinogen. This is key to acknowledge, as it indicates that it is not just the cells themselves that can be affected by changes to O_2_ concentrations, but also key components of an assay, such as matrix proteins, and thus experimental data from these experiments need to appreciate all factors to be fully understood.

## Figures and Tables

**Figure 1 biomolecules-15-00501-f001:**
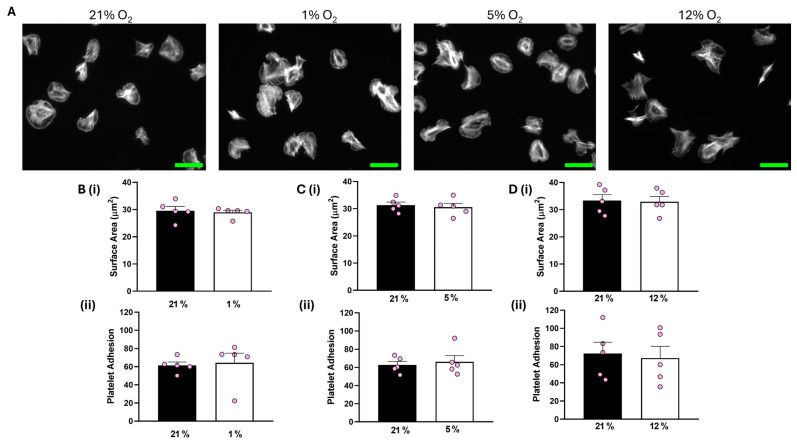
Platelet spreading on collagen is unchanged by O_2_ concentration. Washed platelets (2 × 10^7^ platelets/mL) were spread on a collagen matrix (100 μg/mL) for 25 min before being fixed with PFA, permeabilized with Triton X-100, and stained with FITC-phalloidin. (**A**) Representative images of platelet spreading on collagen at 21%, 1%, 5%, and 12% O_2_ with scale bar showing 10 μm. (**B**) Graphs showing (i) platelet surface area and (ii) average platelet adhesion at 1 % O_2_. (**C**) Graphs showing (i) platelet surface area and (ii) platelet adhesion at 5% O_2_ and (**D**) graphs showing (i) average surface area and (ii) platelet adhesion at 12% O_2_. Data are presented as mean ± SEM. *n* = 5. Statistical analysis was calculated using a paired *t*-test.

**Figure 2 biomolecules-15-00501-f002:**
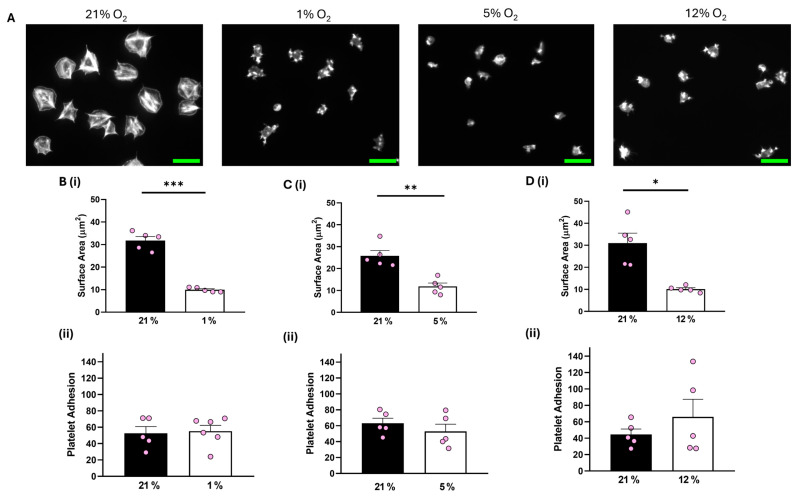
Platelet spreading on fibrinogen is reduced at physiologically relevant O_2_ levels. Washed platelets (2 × 10^7^ platelets/mL) were spread on a fibrinogen matrix (100 μg/mL) for 25 min before being fixed with PFA, permeabilized with Triton X-100, and stained with FITC-phalloidin. (**A**) Representative images of platelet spreading on fibrinogen at 21%, 1%, 5%, and 12% O_2_ with scale bar showing 10 μm. (**B**) Graphs showing (i) platelet surface area and (ii) average platelet adhesion at 1% O_2_, (**C**) graphs showing (i) platelet surface area and (ii) platelet adhesion at 5% O_2_, and (**D**) graphs showing (i) average surface area and (ii) platelet adhesion at 12% O_2_. Data are presented as mean ± SEM. *n* = 5. Statistical analysis was calculated using a paired *t*-test. * *p* < 0.05, ** *p* < 0.01, *** *p* < 0.001.

**Figure 3 biomolecules-15-00501-f003:**
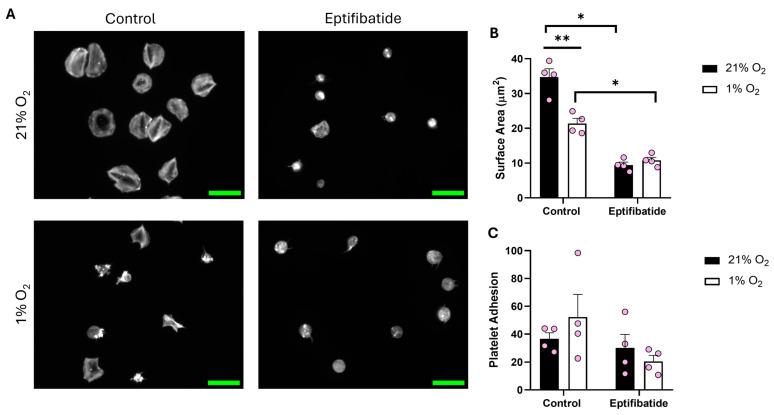
Platelet spreading on fibrin is reduced in hypoxic O_2_ levels. Washed platelets (2 × 10^7^ platelets/mL) were preincubated with eptifibatide (9 μM) for 2 min before spreading on fibrin (100 µg/mL) for 25 min before being fixed with PFA, permeabilized with Triton X-100, and stained with FITC-phalloidin. (**A**) Representative images of platelets treated with zinc, thrombin, and the control at 21% and 1% O_2,_ with scale bar showing 10 μm. Graphs showing (**B**) average platelet surface area and (**C**) platelet adhesion. Data are presented as mean ± SEM. *n* = 4. Statistical analysis was calculated using repeated measures two-way ANOVA with a Tukey’s post hoc test. * *p* < 0.05, ** *p* < 0.01.

**Figure 4 biomolecules-15-00501-f004:**
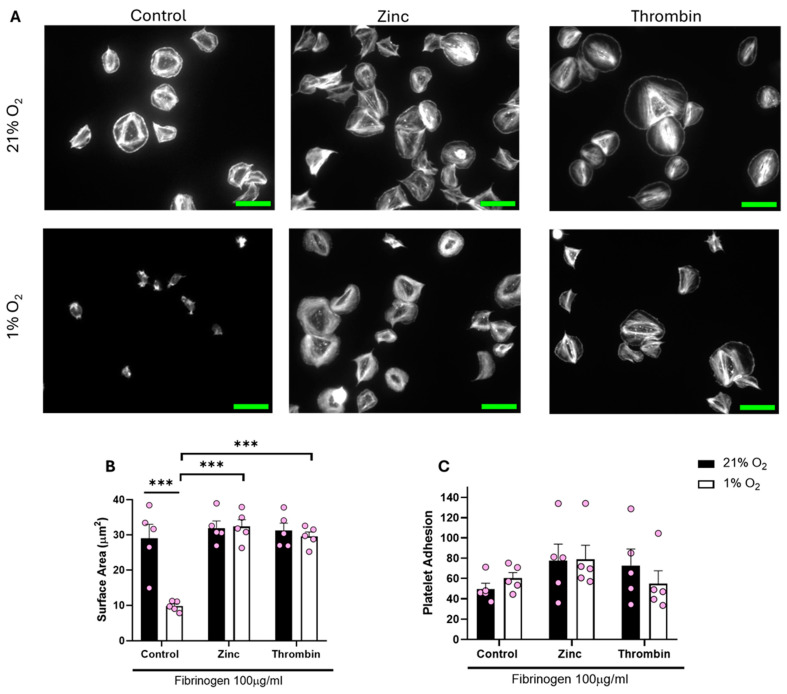
Platelet spreading on fibrinogen at hypoxic O_2_ levels is restored following pre-treatment with zinc and thrombin. Washed platelets (2 × 10^7^ platelets/mL) were preincubated with zinc (100 μM) and thrombin (0.1 U/mL) for 2 min before spreading on fibrinogen (100 µg/mL) for 25 min before being fixed with PFA, permeabilized with Triton X-100, and stained with FITC-phalloidin. (**A**) Representative images of platelets treated with zinc, thrombin, and the control at 21% and 1% O_2,_ with scale bar showing 10 μm. Graphs showing (**B**) average platelet surface area and (**C**) platelet adhesion. Data are presented as mean ± SEM. *n* = 5 Statistical analysis was calculated using repeated measures two-way ANOVA with a Tukey’s post hoc test. *** *p* < 0.001.

**Figure 5 biomolecules-15-00501-f005:**
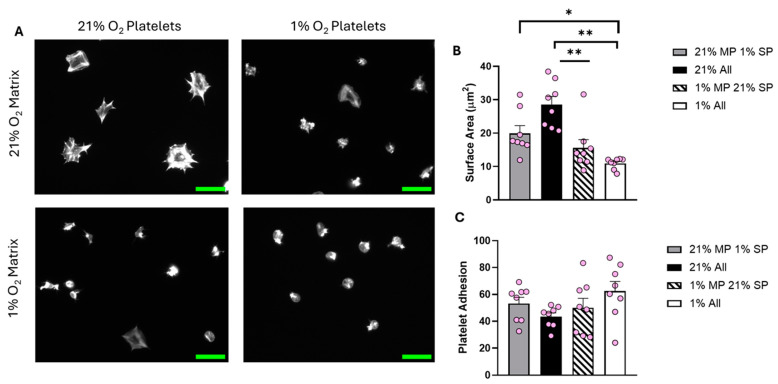
The O_2_ concentration at which the fibrinogen matrix is prepared plays an important role in determining platelet spreading. Fibrinogen matrices were prepared at both 21% and 1% O_2_, platelet spreading was then conducted using washed platelets kept at 21% and 1% O_2_. Washed platelets (2 × 10^7^ platelets/mL) were spread on 100 μg/mL fibrinogen for 25 min before being fixed with PFA, permeabilized with Triton X-100, and stained with FITC-phalloidin. (**A**) Representative images of platelet spreading at each condition, with a scale bar showing 10 μm. Graphs show (**B**) average platelet surface area and (**C**) platelet adhesion. Data are presented as mean ± SEM. *n* = 8. Statistical analysis was calculated using repeated-measures one-way ANOVA with a Tukey’s post hoc test. * *p* < 0.05, ** *p* < 0.01. Matrix prep (MP), spread platelets (SP).

## Data Availability

The original contributions presented in the study are included in the article/Appendix A, further inquiries can be directed to the corresponding author/s.

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
