# Peer review of "Oxygen Concentration Plays a Critical Role in Fibrinogen-Mediated Platelet Activation via Inactivation of αIIbβ3 and Modulation of Fibrinogen"

_biomolecules, 2025, doi:10.3390/biom15040501_

Round 1

Reviewer 1 Report

Comments and Suggestions for Authors

This study presents a detailed assessment of the impact different oxygen concentrations have on platelet spreading on a fibrinogen substrate.  This supports previous literature and extends the work to includes additional physiologically relevant oxygen concentrations.    

The introduction and/or discussion would benefit from a discussion and justification regarding the use of zinc.

Are there any changes in pH caused by the different oxygen concentrations? If so, could this contribute to the observations observed?

Ln 264  The discussion suggests that the preparation of collagen in different O2 concentration does not impact platelet spreading.  Please clarify the statement to ensure that it refers to the platelets incubated at different O2 concentrations prior to spreading.  Was any effect seen with how platelets interact with collagen prepared at 1%?

In the discussion is it suggest that other signalling pathways may be impacted by different concentrations of O2. – A discussion of what these would be beneficial?

A discussion as to why preparation of the fibrinogen substrate under hypoxic conditions results in reduced platelet spreading would be beneficial.

Why was the 3ug/ml fibrinogen coating incubated overnight at 4oC whereas the other coverslips were coated for 1h at room temperature?  What is the difference in the amount of fibrinogen deposited on the coverslip when compared to 100ug/ml?

In supplementary Figure 8.  One of the conditions in the representative trace has a positive population.  It is difficult to determine which condition this is in the representative, is this a common finding?  It would be beneficial to present the data as % positive in addition to MFI, and the outcome discussed.

Can the methodology be clarified?  What was the time between the blood draw and the spreading assay? Please include an overview of any randomisation and/or blinding that was performed.

Why is an unpaired t-test used for the analysis?  If the same donor is used for each experiment, the donor represents the same subject, and a paired T-test would be appropriate, please clarify your approach?

Do the outcomes seen with platelet spreading also seen with platelet aggregation?

What % (v/v) of ACD was used to collect the blood into?

Ln26 Units: this should be µm

Ln 19 it’s to its

Ln 61 venal – change to venous

GP1b – GPIb

Can rpm be changed to g or the centrifuge and rotor included.

Formatting of aIIbB3 (ΑiIbβ3) in reference 6, symbols needed ln 215 aIIbb3

Supplemental Figure 1 – how was platelet oxygen concentration determined?

Figure 1: Can the y-axis in each panel be presented with the same range?

What is used to balance the gas differences in the chamber?

100 mM Zinc was used, how does this compare to physiological levels?

Figure S2 and S3 The figures mentions collagen but the figure is about fibrinogen

Reviewer 2 Report

Comments and Suggestions for Authors

In this study Leonard et al. report the effects of reducing oxygen levels on platelet spreading on fibrinogen in this simple but effective study. The findings that low O2 reduces integrin induced activation through both lowering αIIbβ3 activity and reducing the effectiveness of the fibrinogen matrix is interesting but the reliance on platelet spreading assays and the use of fibrinogen monolayers brings questions to how physiologically relevant this data is. In addition, currently not enough data is presented to rule out the role other platelet receptors may play such as GPVI. If this issue can be addressed, it will greatly improve the study. Specific comments are below:

1)      There are two major platelet receptors for collagen, αIIβI and GPVI, and therefore it is hard to determine if the low oxygen levels is effecting one of these receptor pathways but it’s being maxed by activation of the other. The authors should consider using more specific collagen peptide agonists in the spreading assays to distinguish the difference between GPVI and αIIβI (Collagen related peptide and GFOGER) .

2)      Further to point 1, fibrinogen has been shown to activate GPVI when immobilised on a surface in spreading assays. It has been shown that αIIbβ3 is critical for adhesion, but GPVI is required for full spreading which could explain why platelet spreading was reduced with no significant impact on platelet adhesion numbers. I recommend the authors use specific GPVI/ αIIbβ3 inhibitors to see how they further effect platelet adhesion in normal and low O2 levels. This will help determine the significance of both GPVI and αIIbβ3 in this process. In addition, measuring the levels of phosphorylated FcRγ chain in spread platelets under normal and low oxygen will highlight if GPVI is in some way responsible for this reduction in platelet spreading.

3)      Can the authors also look at spreading on fibrin? Fibrin is known to bind to a distinct binding site on αIIbβ3 to fibrinogen and this would help determine if the low O2 is affecting αIIbβ3 pre-activation or fibrinogen binding/coating.

4)      Focussing on platelet spreading is somewhat limiting the study. Could the authors extend these experiments and look at the effects of low oxygen levels on platelet aggregation in response to fibrinogen? This seems like an easy and obvious extension of the platelet spreading studies. This would also help answer my previous point as GPVI is not activated by soluble fibrinogen.

5)      Can the authors measure/ image the fibrinogen surface under normal and low O2 levels? This will provide more insight into how O2 effects the fibrinogen coating.

Minor comments

1)      The authors should be consistent in using RCF and not RPM when referencing spins.

2)      Figure 1 B, C and D shows different values for 21 % O2 between panels and I am unsure why this is needed. Would it not be better to combine all the data points for 21 % and then show 21 %, 12 %, 5 % and 1 % on one graph instead of three separate ones? The same for figure 2.

3)      I cannot find mention for how the oxygen concentrations in figure S1 were calculated in the methods section.

Round 2

Reviewer 1 Report

Comments and Suggestions for Authors

The authors have addressed my comments and the manuscript is much improved. 

Reviewer 2 Report

Comments and Suggestions for Authors

I am happy with the changes and additions that have been made. Most of my queries have been answered.